☮ | Open Peer Review | Epidemiology | Research Article

# A MALDI-TOF MS-based multiple detection panel of drug resistance-associated multiple single-nucleotide polymorphisms in *Candida tropicalis*

Feifei Wan,[1] Min Zhang,[1] Jian Guo,[1] Huiping Lin,[1] Xiaoguang Zhou,[2] Lixin Wang,[2] Wenjuan Wu[1]

**ABSTRACT**    *Candida tropicalis* is one of the main causes of invasive candidiasis. Rapid identification of antifungal resistance is crucial for selection of an appropriate antifungal to improve patient outcomes. Mutations at specific loci are strongly correlated with resistance to antifungal agents. In this study, we developed a multi-single-nucleotide polymorphism (SNP) panel to accurately identify 36 mutation sites across seven genes of *C. tropicalis* that are associated with resistance to azoles and/or echinocandins. Ten isolates were selected to test repeatability, and another 20 isolates of *C. tropicalis* were selected to validate consistency. Intra-assay and inter-assay repeatability of the panel was 100%, with the loci accuracy being 99.44% (716 of 720). Furthermore, 109 isolates were examined for clinical research, and the most commonly detected mutations were G751A and A866T of *UPC2*, A491T of *TAC1*, and A395T and C461T of *ERG11*. The G751A and A866T mutations of *UPC2* as well as the A395T and C461T mutations of *ERG11* co-existed. The SNP panel enables identification of specific mutations at critical sites of drug-resistant strains to facilitate the rapid selection of appropriate antifungal agents and efficient monitoring of the regional epidemiological trends of resistance of *C. tropicalis*.

**IMPORTANCE**    *C. tropicalis* infections pose a growing global public health challenge, with mortality rates approaching 40%. *C. tropicalis* is one of the top four *Candida* spp. responsible for candidiasis, particularly in the Asia-Pacific region and Latin America, notably affecting patients with neutropenia and malignancies. The azole resistance rate of *C. tropicalis* ranges from 0% to 30%. Between 2009 and 2018, the China Hospital Invasive Fungal Surveillance Network reported an increase in fluconazole and voriconazole resistance from 5.7% to ~30%. Although resistance to echinocandins and amphotericin B remains low, multi-resistance to echinocandins and azoles has been observed. Current methods for detecting drug resistance are limited by the long turnaround time of antifungal susceptibility testing, low throughput of Sanger sequence to target resistance mutations, complex data analysis, and high costs of second-generation sequencing. We developed and validated a rapid, high-throughput, and cost-effective panel to detect and monitor drug-resistance mutations of *C. tropicalis*.

**KEYWORDS**    *Candida tropicalis*, matrix-assisted laser desorption/ionization-time of flight mass spectrometry, single-nucleotide polymorphism

I nvasive candidiasis (IC) refers to systemic fungal infections caused by *Candida* spp., including candidemia and deep-seated infection (1). Mortality of IC can be as high as 40%, especially in immunocompromised, profound neutropenic, and intensive care unit (ICU) patients. Approximately 50% of candidaemia cases occur in the ICU (1, 2). While more than 15 distinct *Candida* spp. can cause human infections, the 6 most clinically

Address correspondence to Wenjuan Wu, wwj1210@126.com.

Feifei Wan and Min Zhang contributed equally to this article. The author order was determined by Wenjuan Wu.

The authors declare no conflict of interest.

See the funding table on p. 10.

relevant are *Candida albicans*, *Candida parapsilosis*, *Pichia kudriavzevii*, *Nakaseomyces glabratus*, *Candida tropicalis*, and *Candida auris* (3). *C. tropicalis* is associated with high morbidity and mortality due to the formation of strong biofilms, high virulence, hospital horizontal transmission, and antifungal resistance (4, 5). Recent evidence indicates an azole resistance rate of 20%–50% for *C. tropicalis* in the Asia-Pacific region, India, Turkey, and Algeria (6–9). The emergence of fluconazole resistance has led to increased echinocandin exposure, with *C. tropicalis* isolates resistant to both azoles and echinocandins being reported in the USA (10, 11), India (7), Taiwan (12), and mainland China (13), although the prevalence remains low (<2%).

Azoles and echinocandins are the first-line therapeutic agents for IC. Azoles targeting 14-alpha-demethylase (Erg11) inhibit the ergosterol biosynthesis pathway in fungal cell membranes, whereas echinocandins bind to the Fks p-subunit of the β-(1,3)-D-glucan synthase complex, thereby blocking synthesis of β-(1,3)-D-glucan (14, 15). Azole resistance can develop through multiple mechanisms, including substitutions in Erg11, which reduce the drug-binding affinity between ergosterol and azoles (16, 17). Additionally, overexpression of Erg11, driven by gain-of-function mutations to the transcriptional activator *UPC2*, can lead to overexpression of drug targets. Azole resistance is also associated with upregulation of drug efflux pumps. Mutations to *TAC1* or *MRR1* lead to the upregulation of adenosine triphosphate-binding cassette transporter *CDR1* and the major facilitator superfamily transporter *MDR1*. Resistance to echinocandins primarily involves mutations to *FKS1* (16).

The broth microdilution method is recommended as the reference method for antifungal susceptibility testing to determine the minimal inhibitory concentration (MIC) by the Clinical and Laboratory Standards Institute and the European Committee on Antimicrobial Susceptibility Testing (18, 19). However, this method is limited by a long turnaround time, tedious procedures, and subjective interpretation of the results. Recent advancements in research on drug resistance mechanisms of *C. tropicalis* have led to the development of molecular biology methods to detect resistance-related mutations, aiming to clarify antifungal resistance. Sanger sequencing is regarded as the gold standard. However, its single-target nature makes it relatively time consuming and costly. Considering the emergence of azole resistance of *C. tropicalis* and increased exposure to echinocandins, developing alternative molecular methods is essential for rapid, reliable, accurate, and cost-effective monitoring of antifungal resistance to optimize treatment outcomes.

Matrix-assisted laser desorption ionization time-of-flight mass spectrometry (MALDI-TOF MS) was first used for detection of single-nucleotide polymorphisms (SNPs) in human genomic DNA in 1998 (20) and more recently has been applied for genotyping of SNPs of human herpesviruses, human papillomavirus, methicillin-resistant *Staphylococcus aureus*, and *Mycoplasma pneumoniae* (21–24). The advantages of MALDI-TOF MS as an alternative molecular biology method for genotyping SNPs include high throughput, real-time analysis, accuracy, precision, and cost-effectiveness compared to other methods.

Here, we report the development of a multi-SNP detection panel based on MALDI-TOF MS to identify the drug-resistant phenotypes and facilitate epidemiological studies of the SNPs of *C. tropicalis* genes associated with antifungal resistance.

## RESULTS

### Repeatability evaluation

The repeatability of the panel was assessed using 10 *C. tropicalis* isolates. Intra-assay repeatability was evaluated using three technical replicates per isolate, and inter-assay repeatability was assessed across two independent experiments. All replicates across both intra-assay and inter-assay conditions demonstrated complete concordance at all loci. As no discrepancies were observed, the panel exhibited 100% repeatability.

Overall, mutations were detected at nine distinct loci of four genes, including T1949C of *FKS1*; A395T, T433C, and C461T of *ERG11*; G751A, A866T, and C1178T of *UPC2*; and

A491T of *TAC1*. Among the tested isolates, four (ECIFIG685, ECIFIG687, ECIFIG1112, and ECIFIG1831) were resistant to both fluconazole and voriconazole, involving SNPs of *ERG11* (A395T and C461T), *UPC2* (G751A and A866T), and *TAC1* (A491T). One additional isolate, resistant to fluconazole and intermediate to voriconazole, carried mutations to *UPC2* (G751A, A866T, and C1178T). An isolate, susceptible-dose dependent to fluconazole and intermediate to voriconazole, harbored the mutation T433C of *ERG3*. In contrast, isolates susceptible to both fluconazole and voriconazole (e.g., ECIFIG31, ECIFIG343, and ECIFIG1642) had fewer or no significant resistance-associated SNPs in *UPC2* (G751A and A866T) or *TAC1* (A491T). Notably, isolates ECIFIG1642 and ECIFIG21032, which were resistant to anidulafungin, lack resistance-associated SNPs of *FKS1*. Other isolates (except strain ECIFIG1112), which were susceptible to all three echinocandins, did not have resistance-associated SNPs of *FKS1* (Table 1).

## Coherence assessment

To evaluate the accuracy of the multiplex SNP detection panel for *C. tropicalis*, 20 *C. tropicalis* isolates were tested across 36 SNP loci, resulting in a total of 720 SNP evaluations. Strains ECIFIG159 and ECIFIG385 demonstrated discordance at a single SNP locus (*ERG11*-395 and *FKS1*-1958, respectively), while strain ECIFIG455 exhibited discordance at two loci (*UPC2*-751 and *UPC2*-787). The remaining 716 loci were found to be concordant with the reference method. The system demonstrated an overall accuracy of 99.44% (716 of 720) (Table 2).

All isolates had at least one mutation. Among the 36 loci examined, mutations were detected at 12 loci across six genes (Table 2). Due to the diploid nature of *C. tropicalis*, both homozygous and heterozygous mutations were detected.

## Clinical performance evaluation

Given the high consistency rate and reliability, this panel was further used to identify mutations of 109 *C. tropicalis* isolates. In total, 14 distinct loci with mutations were identified, which were mainly concentrated at loci *UPC2*-751 (65.14%, 71 of 109), *UPC2*-866 (64.22%, 70 of 109), *TAC1*-491 (62.39%, 68 of 109), *ERG11*-461 (38.53%, 42 of 109), and *ERG11*-395 (36.70%, 40 of 109). Additionally, co-existence of mutations at the loci *UPC2*-751 and *UPC2*-866, as well as *ERG11*-395 and *ERG11*-461, was frequent. Mutations at *ERG11*-395 and *ERG11*-461 were exclusive to azole-resistant *C. tropicalis*, whereas mutations at *UPC2*-751, *UPC2*-866, and *TAC1*-491 occurred in isolates resistant and not resistant to azoles. Among isolates with a mutation at the *ERG11*-461 locus, 95.24% also had a mutation at the *ERG11*-395 locus; 90.48% had a mutation at the *TAC1*-491 locus; and 92.86% had mutations at the *UPC2*-751 and *UPC2*-866 loci. Among isolates with a mutation at the *UPC2*-751 locus, 95.77% also had a mutation at the

**TABLE 1** SNPs of 10 *C. tropicalis* isolates identified by MALDI-TOF MS and selected to evaluate repeatability[a]

| Strain | Year of isolation | FKS1 1949 T→C | ERG11 | | | UPC2 | | | | TAC1 491 A→T | Susceptibility to echinocandins and azoles | | | | |
|---|---|---|---|---|---|---|---|---|---|---|---|---|---|---|---|
| | | | 395 A→T | 433 T→C | 461 C→T | 751 G→A | 866 A→T | 1178 C→T | 1712 T→A | | ANI | CAS | MICA | FLU | VOR |
| ECIFIG31 | 2017 | T | A | T | C | G, A | A, T | C | T | T | S | S | S | S | S |
| ECIFIG343 | 2018 | T | A | T | C | G, A | A, T | C | T | A | S | S | S | S | S |
| ECIFIG685 | 2018 | T | A, T | T | C, T | G | T | C | T | T | S | S | S | R | R |
| ECIFIG687 | 2018 | T | A, T | T | C, T | A | T | C | T | T | S | S | S | R | R |
| ECIFIG1112 | 2019 | C | A, T | T | C, T | A | T | C | A | A, T | S | S | S | R | R |
| ECIFIG1448 | 2019 | T | A | T | C | G | A | C | T | A | S | S | S | S | S |
| ECIFIG1642 | 2020 | T | A | T | C | G | A | C | T | A, T | R | S | S | S | S |
| ECIFIG1831 | 2020 | T | A, T | T | C, T | A | T | C | T | A, T | S | S | S | R | R |
| ECIFIG21032 | 2021 | T | A | T | C | G, A | T | T | T | A | R | I | R | R | I |
| ECIFIG21340 | 2021 | T | A | C | C | G | A | C | T | A | S | S | S | SDD | I |

[a]Abbreviations: ANI, anidulafungin; CAS, caspofungin; FLU, fluconazole; I, intermediate; MICA, micafungin; R, resistant; S, susceptible; SDD, susceptible-dose dependent; VOR, voriconazole.

**TABLE 2** SNPs of 20 *C. tropicalis* isolates identified by MALDI-TOF MS and selected for coherence assessment

| Strain | Year of isolation | FKS1 | | ERG11 | | | | UPC2 | | | TAC1 | MDR1 | MRR1 |
|---|---|---|---|---|---|---|---|---|---|---|---|---|---|
| | | 1958 | 1960 | 395[a] | 433 | 461[a] | 1286 | 751 | 787 | 866 | 491 | 227 | 1939 |
| | | T→C | TC→CC | A→T | T→C | C→T | C→T | G→A | G→A | A→T | A→T | T→C | G→T |
| ECIFIG27 | 2017 | T | TC | A | T | C | C | G | G | A | A,T | T | G |
| ECIFIG113 | 2017 | T | TC | A | T | C | C | G | G | A | A,T | T | G |
| ECIFIG158 | 2017 | T | TC | T | T | C,T | C | G,A | G | A,T | A,T | T | G |
| ECIFIG159 | 2017 | T | TC | A (A,T) | T | C,T | C | G,A | G | A,T | A,T | T | G |
| ECIFIG189 | 2017 | T | TC | A | T | C | C | G | G | A | A,T | T | G |
| ECIFIG196 | 2017 | T | TC | A | T | C | C | A | G | T | A | T | G |
| ECIFIG214 | 2017 | T | TC | A | T | C | C | G | A | A | A,T | T | G,T |
| ECIFIG260 | 2018 | T | TC | A | T | C | C,T | A | G | T | A,T | T | G |
| ECIFIG354 | 2018 | T | TC | A | T | C | C | A | G | T | A | T | G |
| ECIFIG358 | 2018 | T | TC | A | T,C | C | C | G,A | G | A,T | A | T | G |
| ECIFIG385 | 2018 | T,G (T) | TC | A | T | C | C | G | G | A | A,T | T | G |
| ECIFIG455 | 2018 | T | TC | A | T | C | C | G (G,A) | G,A (G) | A,T | A | T | G |
| ECIFIG760 | 2018 | T | TC | A | T | C | C | G,A | G,A | A,T | A,T | T | G |
| ECIFIG818 | 2018 | T | TC | A | C | C | C | G | G | A | A | T | G |
| ECIFIG833 | 2018 | T | TC | A | T | C | C | G | G | A | A,T | T | G |
| ECIFIG844 | 2018 | T | TC | A | T | C | C | G | G | A | A,T | T,C | G |
| ECIFIG885 | 2019 | T | TC | T | T | T | C | G,A | G | A,T | A,T | T | G |
| ECIFIG889 | 2019 | T | TC | T | T | T | C | G,A | G | A,T | A,T | T | G |
| ECIFIG909 | 2019 | T | TC | T | T | T | C | G,A | G | A,T | A,T | T | G |
| ECIFIG1521 | 2020 | T | CC | T | T | T | C | G,A | G | A,T | A,T | T | G |

[a]SNPs associated with antifungal resistance (gene editing level). Mutations detected by MALDI-TOF MS are underscored. When the results detected by MALDI-TOF MS were inconsistent with the Sanger sequence, the latter results are indicated in parentheses.

*UPC2*-866 locus; 60.56% had a mutation at the *TAC1*-491 locus; 53.52% had a mutation at the *ERG11*-395 locus; and 54.92% had a mutation at the *ERG11*-461 locus (Fig. 1).

The mutations at the loci *ERG11*-395, *ERG11*-461, and *ERG11*-769 were related to azole resistance, as confirmed by gene editing. In this study, all isolates with mutations at these loci exhibited azole resistance. For isolates with mutations at both the *ERG11*-395 and *ERG11*-461 loci, 90% (36 of 40) were highly resistant to fluconazole (MIC ≥128 g/L) and voriconazole (MIC ≥8 g/L). However, not all azole-resistant isolates carried these three mutations, as some carried mutations at other loci associated with azole resistance, such as *UPC2*-1029 (2.75%, 3 of 109), *UPC2*-751 (65.14%, 71 of 109), *TAC1*-491 (62.39%, 68 of 109), *ERG11*-433 (1.83%, 2 of 109), and *UPC2*-866 (64.22%, 70 of 109). Additionally, isolates with SNPs at the *UPC2*-1029 and *ERG11*-433 loci were not susceptible to fluconazole and voriconazole. However, these SNPs were absent in other isolates susceptible to fluconazole and voriconazole. Among the strains with mutations at both the *UPC2*-751 and *UPC2*-866 loci, 70.58% (48 of 68) were resistant to fluconazole and voriconazole. Notably, ECIFIG21010269 did not have any of the mutations and was sensitive to voriconazole but resistant to fluconazole (MIC = 8 g/L).

The SNPs at loci 1949 (5.50%, 6 of 109) and 1960 (0.91%, 1 of 109) of *FKS1* were associated with resistance to echinocandins. The SNP at locus 1949 of *FKS1* was identified in two isolates that were resistance to anidulafungin and micafungin, while the remaining four isolates were sensitive to echinocandins. The SNP at locus 1960 of *FKS1* was identified in one isolate, which exhibited intermediate resistance to caspofungi. Additionally, four isolates with mutations were not susceptible to echinocandins.

## DISCUSSION

*C. tropicalis* infections and the emergence of drug resistance present a significant threat to human health. There is an urgent need to develop rapid and accurate methods to detect antifungal agent sensitivity. Currently, microbroth dilution is recommended as the reference method forantifungal susceptibility testing, but this method

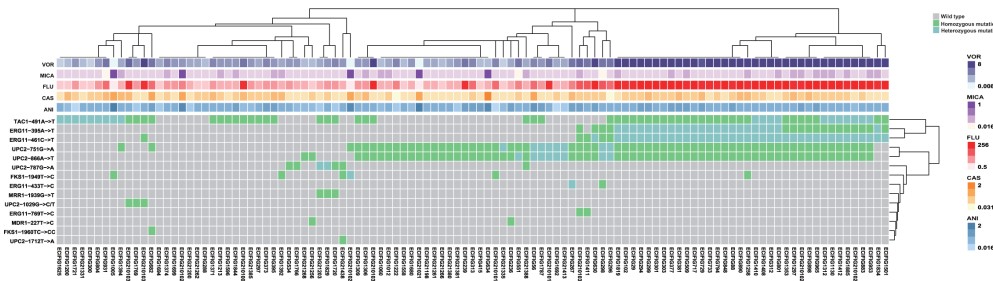

**FIG 1** Cluster heat map of mutations of 109 *C. tropicalis* isolates. The heat map was generated based on the mutations at 14 loci of 109 *C. tropicalis* isolates. Gray indicates wild type; green indicates homozygous mutations; and blue indicates heterozygous mutations. The horizontal coordinates of the heat map represent the isolates, and the vertical coordinates represent the gene loci. The color annotation bar at the top demonstrates the antifungal agent susceptibility of each isolate of *C. tropicalis*. The MICs are listed in in Table S1.

requires a long turnaround time, making it impractical for situations that require quick results (25). Current molecular biology detection methods to detect drug resistance include polymerase chain reaction (PCR), PCR-restriction fragment length polymorphism analysis, PCR-single-strand conformation polymorphism analysis, TaqMan probe technology, gene chip technology, and next-generation sequencing (26–29). However, PCR detection sensitivity for SNP depends on the equipment, while next-generation sequencing is time consuming, complex, and requires expensive equipment. The resistance mechanisms of SNPs at specific loci are often multi-factorial, involving multiple SNP loci and different regulatory pathways (30). This complexity means that a single biomarker may not be sufficient to fully capture the drug resistance profile. Therefore, a combination of potential biomarkers is recommended to determine the antifungal susceptibility of *C. tropicalis* (31, 32). Multiplex PCR with single base pair extension is a useful technique for detection of more than 10 SNP sites and can be adopted for genotyping and identification of multiple microorganisms (33).

The developed panel based on MALDI-TOF MS combined with multiplex PCR and single base pair extension can accurately identify SNPs of *C. tropicalis* associated with resistance to azoles and echinocandins within 6–8 hours. The panel includes 36 SNP sites, including 9 related to azole resistance, as confirmed by gene editing. The accuracy of the panel was 99.44%, with intra-assay and inter-assay repeatability achieving 100%, demonstrating that the panel has potential for clinical detection of drug-resistant gene mutations in *C. tropicalis*.

The mutations of all 109 *C. tropicalis* isolates collected from eastern China were mainly concentrated at five sites: *UPC2*-751, *UPC2*-866, *TAC1*-491, *ERG11*-395, and *ERG11*-461. The co-occurrence of mutations at *UPC2*-751 and *UPC2*-866, as well as *ERG11*-395 and *ERG11*-461, was remarkably high. The mutations at the *ERG11*-395 and *ERG11*-461 loci were exclusive to azole-resistant isolates. In total, 14 SNP loci were detected in this clinical study, which included 3 (i.e., *ERG11*-395, *ERG11*-461, and *ERG11*-769) previously confirmed by gene editing to confer azole resistance. The results of global antifungal monitoring showed that azole resistance of *C. tropicalis* is mainly due to the A395T substitution of *ERG11* and overexpression of Erg11 (34, 35). This drug-resistant mutation is concentrated in Thailand (36). A study conducted in China indicated that A395T and C461T were the most commonly reported non-homologous mutations to *ERG11* related to azole resistance. The independent functions of these two sites were also investigated. The *ERG11* A395T substitution, but not the C461T mutation, was associated with azole resistance. However, these mutations often appear in conjunction. To date, there has been no report of an isolated C461T mutation (37). In this study and in a previous research, the A395T, C461T, and T769C substitutions of *ERG11* were found only in azole-resistant strains. However, in contrast to previous studies, the C461T substitution was independently present in two azole-resistant isolates (i.e.,

ECIFIG1411 and EC21010386), suggesting the possibility of other resistance mechanisms in these two strains. Isolate ECIFIG1411, but not EC21010386, carried the *ERG11*-T769C mutation, which has been associated with azole resistance. However, EC21010386 had the *UPC2*-T1029C mutation, which was found in one isolate that was resistant to azoles and another isolate that was susceptible to fluconazole but intermediate to voriconazole. These results indicate that the *UPC2*-T1029C mutation is probably linked to azole resistance. Also, some azole-resistant isolates did not carry SNPs at the *ERG11*-395, *ERG11*-461, and *ERG11*-769 loci but rather other azole-associated mutations. The mutation at locus *ERG11*-433 was unique to the azole-resistant isolates ECIFIG267 and ECIFIG298 and was not carried by any other azole-sensitive isolate, suggesting that this locus is important for azole resistance. In this study, mutations at position 1949 of the *FKS1* gene were identified in two isolates that exhibited resistance to anidulafungin and micafungin, consistent with previous findings that *FKS1* mutations led to echinocandin resistance (36). Additionally, a mutation at position 1960 of the *FKS1* gene was detected in one isolate with intermediate to caspofungin, a mutation that has been associated with resistance in other studies (11). Notably, four isolates with mutations at position 1949 of the *FKS1* gene were not resistant to echinocandins, suggesting that this specific mutation might not drastically alter the conformation of enzymes to confer full resistance but rather might cause only partial structural changes, leading to decreased susceptibility but not complete resistance. Also, echinocandin resistance may not solely depend on *FKS1* mutations but could involve other resistance mechanisms (38). Moreover, four isolates exhibited reduced susceptibility to echinocandins but had no *FKS1* mutations, indicating the possibility of mutations at other sites of the *FKS1* gene, cellular stress response, or the upregulation of multi-drug transporters contributing to resistance (39).

MALDI-TOF MS provides highly automated processes and can be applied for multiplex assays, reducing time and cost while increasing sample throughput. The established MALDI-TOF MS-based detection panel is both practical and feasible for rapid and large-scale scanning of mutations of *C. tropicalis* related to drug resistance. Epidemiological monitoring of resistance to antifungal agents is especially important in regions where echinocandin-resistant strains have become increasingly prevalent. However, the study has some limitations. During the performance validation of the panel, it was discovered that due to base mismatches, the panel may incorrectly identify non-mutation as heterozygous mutation and vice versa, which could result in the misinterpretation of results. Given the diverse mechanisms of drug resistance in *C. tropicalis*, this panel currently includes only the most significant mutations, potentially overlooking less common or emerging resistance mechanisms. Additionally, the panel is designed to detect mutations from pure cultures of *C. tropicalis*, which may limit direct application in clinical settings. Further efforts will be required to adapt this method for detecting SNPs directly from clinical specimens, such as blood or tissue samples, to enhance its clinical utility.

## MATERIALS AND METHODS

### Isolates

*C. tropicalis* isolates were collected from patients with invasive fungal infections in the Eastern China Invasive Fungi Infection Group (ECIFIG). Patients with invasive fungal disease were identified in accordance with the definitions of definitive diagnoses of invasive fungal diseases by the European Organization for Research and Treatment of Cancer and the Mycoses Study Group Education and Research Consortium (40). Each isolate was identified by MALDI-TOF MS (Zybio, Inc., Chongqing, China). Antifungal susceptibility of the *C. tropicalis* isolates was determined using the Sensititre YeastOne panel (Thermo Fisher Scientific, Waltham, MA, USA), which has been verified against the M27-Ed4, "Reference Method for Broth Dilution Antifungal Susceptibility Testing of Yeasts; Fourth Informational Supplement," and the European Committee on

Antimicrobial Susceptibility Testing of yeasts (v.7.3.1 valid from 15 January 2017 to 22 April 2020) (41). The susceptibilities of the *C. tropicalis* isolates to antifungal agents are shown in Table S1. To evaluate the repeatability of the method, 10 isolates of *C. tropicalis* from different years and with varying antifungal resistance phenotypes were selected (Table S1), ensuring a balanced representation of resistance profiles. Additionally, 20 isolates were chosen to assess consistency, also derived from diverse years and resistance phenotypes, maximizing the detection of the included SNP sites. To evaluate the clinical performance of the method, all *C. tropicalis* isolates collected between 2017 and 2021 by the ECIFIG were included.

## Target gene selection for antifungal susceptibility

The following seven genes of *C. tropicalis* isolate MYA-3404 associated with antifungal resistance were selected from the National Center for Biotechnology Information database for detection of SNPs: *ERG11* (gene accession number M23673), *FKS1* (gene accession number EU676168.2), *TAC1* (gene accession number XM_002550963.1), *MRR1* (gene accession number XM_002547926.1), *ERG3* (gene accession number XM_002550136), *UPC2* (gene accession number NW_003020056.1), and *MDR1* (gene accession number XM_002548069). In total, 36 SNPs of these genes were selected, which included 9 SNPs previously associated with antifungal resistance through gene editing and 27 others related to antifungal resistance of *C. tropicalis* (Table 3).

## Design of PCR amplification primers and mass probes

Multiplex PCR primers and mass probes (Table S2) were designed using BatchPrimer3 v.1.0 software (http://wheat.pw.usda.gov/demos/BatchPrimer3/). Also, 15 sets of PCR primers were designed to amplify the target sites for multiplex PCR. The lengths of the primers ranged from 16 to 25 bp, with a 10-bp fixed sequence (ACGTTGGATG) added to the 5′ end of each primer. The lengths of the mass probes were between 14 and 27 bp. The molecular weight was designed to be 4–9 kDa with a minimal difference of 16 Da among the probes.

## PCR-MALDI-TOF MS and data analysis

The operation procedure mainly included DNA extraction, multiplex PCR amplification, mass probe extension (MPE), MALDI–TOF MS data acquisition, and QuanSNP analysis. The study protocol was largely similar to a previously reported method but with minor modifications (24). Total DNA was extracted from all pure cultures of *C. tropicalis* using the phenol-chloroform extraction method, following standard protocols with minor modifications. Briefly, colonies were scraped from Sabouraud dextrose agar, re-suspended in 500 µL of phosphate-buffered saline, and pelleted by centrifugation. The pellet was re-suspended in 500 µL of lysis buffer (50-mM Tris-HCl, pH 8.0, 50-mM ethylenediaminetetraacetic acid, and 1% sodium dodecyl sulfate), and 0.2-µm glass beads were added. Cells were lysed using the FastPrep-24 5G Bead Beating Grinder and Lysis System (MP Biomedicals, Santa Ana, CA, USA). The lysate was extracted with phenol:chloroform: alcohol (25:24:1, vol/vol) and centrifuged. The aqueous phase was transferred and re-extracted with chloroform. DNA was precipitated with absolute ethanol, washed with 75% ethanol, air-dried, and re-suspended in 50 µL of TE buffer (10 mM Tris-Cl, pH 8.0, 1-mM EDTA). DNA concentration and purity were measured using a Quawell Q5000 Micro-Volume UV-Vis Spectrophotometer (Quawell Technology, Sunnyvale, CA, USA). The DNA was considered sufficiently pure at A260/280 = 1.6–2.0 and A260/230 >1. Nucleic acid-free water was used as a blank control. The regions surrounding the target genes were amplified by multiplex PCR. Each reaction volume included 2 µL of DNA (5–10 ng/mL), 2 µL of buffer (Intelligene Biosystems Co., Ltd., Qingdao, China), and 1 µL of primers. The amplification protocol included an initial denaturation step at 95℃ for 15 min followed by 30 cycles at 95℃ for 15 s, 59℃ for 30 s, 72℃ for 30 s, and 60℃ for 10 min, and then cooled to 4℃. To eliminate free deoxynucleotide triphosphates, the PCR products were digested with 2 mL of shrimp alkaline phosphatase at 37℃ for

**TABLE 3** List of 36 SNPs of seven *C. tropicalis* genes associated with resistance to azoles and echinocandins

| Genes | SNPs | Amino acid substitutions | Phenotype | Confirmed Level | References |
|---|---|---|---|---|---|
| FKS1 | T1960C | S654P | Echinocandin resistant | Epidemiology | (42, 43) |
| | T1958G | L653W | Echinocandin resistant | Epidemiology | (11, 44) |
| | T1949C | F650S | Echinocandin resistant | Epidemiology | (45) |
| *ERG3* | C774T | S258F | Azole resistant | Gene editing | (46) |
| | C773T | S258F | Azole resistant | Epidemiology | (47) |
| *MRR1* | G1939T | A647S | Azole resistant | Epidemiology | (48) |
| *TAC1* | A491T | N164I | Azole resistant | Epidemiology | (48) |
| *MDR1* | T227C | V76A | Azole resistant | Epidemiology | (36) |
| *ERG11* | A395T | Y132F | Azole resistant | Gene editing | (37) |
| | C461T | S154F | Azole resistant | gene editing | (49, 50) |
| | T433C | F145L | Azole resistant | Epidemiology | (51) |
| | C1286T | S429F | Azole resistant | Epidemiology | (51, 52) |
| | A427C | K143X | Azole resistant | Epidemiology | (53) |
| | A428G | K143R | Azole resistant | Gene editing | (49, 50) |
| | T1334A | D445V | Azole resistant | Epidemiology | (37) |
| | A1172- | Δ126aa | Azole resistant | Epidemiology | (46, 47) |
| | Δ132 bp (NA 824–955) | Δ44aa (AA276-319) | Azole resistant | Gene editing | (46) |
| | Δ132 bp (NA 824–955) | D275V | Azole resistant | Epidemiology | (54) |
| | G1391A | G464D | Azole resistant | Epidemiology | (54) |
| | T374C | V125A | Azole resistant | Gene editing | (54) |
| | T769C | Y257H | Azole resistant | Gene editing | (54) |
| | G1390A | G464S | Azole resistant | Gene editing | (55) |
| | C997A | L333I | Azole resistant | Epidemiology | This study |
| | G1032C/T | K344N | Azole resistant | Epidemiology | (55) |
| | G1084A | V362M | Azole resistant | Epidemiology | (37, 54) |
| | T1086G | V362M | Azole resistant | Epidemiology | (46) |
| *UPC2* | A1020T | Q340H | Azole resistant | Epidemiology | (54) |
| | A1141T | T381S | Azole resistant | Epidemiology | (54) |
| | T1712A | F571Y | Azole suscetible dose dependent | Epidemiology | (48) |
| | G787A | A263T | Azole resistant | Epidemiology | (54) |
| | G751A | A251T | Azole resistant | Epidemiology | (54) |
| | A866T | Q289L | Azole resistant | Epidemiology | (48) |
| | G889T | A297S | Azole resistant | Epidemiology | (54) |
| | G1029C/T | L343F | Azole resistant | Epidemiology | (48) |
| | C1178T | T393I | Azole resistant | Gene editing | (54, 56) |
| | C560T | S187L | Azole resistant | Epidemiology | (48) |

40 min and 85°C for 5 min, and then cooled to 4°C. Finally, the digested PCR products were mixed with 4 mL of MPE (1 µL of enzyme-linked dideoxynucleotide triphosphate, 1.4 µL of MPE buffer, 0.6 µL of enzyme, and 1 µL of MPE primers) and heated to 95°C for 30 s, followed by five cycles at 95°C for 5 s and 52°C 5 s, and 40 cycles at 80°C for 5 s and 72°C for 3 min, and then cooled to 4°C. After salt purification, the supernatant was purified with a mixture of 3-hydroxypyridine-2-carboxylic acid (1:1). Then, 1 µL of the mixture was spotted on the target plate. After crystallization, the samples were tested. The MALDI-TOF MS data were acquired by QuanNUA (Intelligene Biosystems Co., Ltd.) and analyzed with a QuanTOF I system (Intelligene Biosystems Co., Ltd.). The system can distinguish different MPE primers and molecular weights after extension of MPE primers and present the peak positions of each primer and extended base on the spectrogram.

## Method establishment and optimization

A MALDI-TOF MS-based method was established for high-throughput detection of mutant loci of *C. tropicalis* associated with resistance to echinocandins and azoles. In total, 36 SNPs of seven genes associated with antifungal resistance were selected (Table 3). Optimal amplification primers and mass probes were designed for these loci. To avoid dimer formation, forward or reverse complementary mass probes were used for different loci in the W1 and W2 reactions, which encompassed 18 and 19 mutated loci, respectively. Two different mass probes for the ERG11-428 locus were used, with each reaction including one of the probes, due to the presence of base mutations in some DNA samples. All multiplex-PCR primer sequences, mass probes, and single-base extensions are shown in Table S2. After optimization, the final concentration of the mass probe most suitable for single base extension was determined (Table S2).

## Repeatability assessment

To evaluate the reproducibility of the assay, inter-assay variability was determined on two different experimental days. Each experimental batch comprised 10 *C. tropicalis* isolates, and the assay procedures were independently repeated for each batch. The inter-assay variabilities were then compared to assess the consistency of measurements across different experimental conditions. Intra-assay variability analysis was performed to evaluate the precision and reliability of the assay within the same experimental batch. Three technical replicates were used for 36 loci of the 10 *C. tropicalis* isolates within a single test batch.

## Evaluation of concordance

Sanger sequencing was used as the gold standard method to sequence seven target genes from 20 *C. tropicalis* isolates for validation. Genomic DNA was extracted from pure cultures using the phenol-chloroform extraction method (as previously described). The primers used for Sanger sequencing are listed in Table S3. The Applied Biosystems T100 Thermal Cycler (Bio-Rad Laboratories, Hercules, CA, USA) was used for conventional PCR setup, and the cycling conditions were one cycle at 95°C for 10 min followed by 40 cycles at 95°C for 30 s, 55°C for 30 s, 72°C for 45 s, and a final extension step at 72°C for 10 min. Subsequently, the same primers were selected for sequencing. Sanger sequencing was outsourced to Sangon Biotech Co., Ltd. (Shanghai, China).

## Statistical analysis

Statistical analysis was conducted with Excel 2021 (Microsoft, Redmond, WA, USA). Sample detection results were obtained from MALDI–TOF MS in Excel format (Intelligene Biosystems, IntelliBio), and then descriptive statistical analysis was conducted.

### ACKNOWLEDGMENTS

We thank the Eastern China Invasive Fungi Infection Group members for their helpful collection of clinical isolates.

This work was supported by grants from the National Natural Science Foundation of China (grant numbers 82172326 and 81971990).

### AUTHOR AFFILIATIONS

[1]Department of Laboratory Medicine, Shanghai East Hospital, School of Medicine, Tongji University, Shanghai, China
[2]Intelligene Biosystems (Qingdao) Co., Ltd, Qingdao, China

### AUTHOR ORCIDs

Feifei Wan  http://orcid.org/0009-0008-6510-6582

Huiping Lin ⬤ https://orcid.org/0000-0003-4437-1631
Wenjuan Wu ⬤ http://orcid.org/0000-0002-5161-3179

## FUNDING

| Funder | Grant(s) | Author(s) |
|--------|----------|-----------|
| MOST \| National Natural Science Foundation of China (NSFC) | 82172326 | Wenjuan Wu |
| MOST \| National Natural Science Foundation of China (NSFC) | 81971990 | Wenjuan Wu |

## ETHICS APPROVAL

The study protocol was approved by the Health Research Ethics Board of Shanghai East Hospital (approval no. 2021-063).

## ADDITIONAL FILES

The following material is available online.

### Supplemental Material

**Tables S1 to S3 (Spectrum00764-24-s0001.docx).** This supplementary material provides a description of the MICs, primer sequences, and mass probes used for the isolates in this study.

### Open Peer Review

**PEER REVIEW HISTORY (review-history.pdf).** An accounting of the reviewer comments and feedback.

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
