## [Reviewer comments · Microbiology Spectrum]

Microbiology Spectrum

A MALDI-TOF MS-based multiple detection panel of drug resistance-associated single nucleotide polymorphisms in *Candida tropicalis*

Feifei Wan, Min Zhang, Jian Guo, Huiping Lin, Xiaoguang Zhou, Lixin Wang, and WenJuan Wu

Corresponding Author(s): WenJuan Wu, East Hospital

Review Timeline:

Submission Date:	March 23, 2024
Editorial Decision:	July 20, 2024
Revision Received:	September 25, 2024
Accepted:	October 31, 2024

Editor: Rita Oladele

Reviewer(s): Disclosure of reviewer identity is with reference to reviewer comments included in decision letter(s). The following individuals involved in review of your submission have agreed to reveal their identity: Bram Spruijtenburg (Reviewer #1)

Transaction Report:

DOI: <https://doi.org/10.1128/spectrum.00764-24>

Re: Spectrum00764-24 (A MALDI-TOF MS-based panel for detection of drug resistance-associated multi-SNPs in *Candida tropicalis*.)

Dear Prof. WenJuan Wu:

Thank you for the privilege of reviewing your work. Below you will find my comments, instructions from the Spectrum editorial office, and the reviewer comments.

Revision Guidelines

Sincerely,
Rita Oladele
Editor
Microbiology Spectrum

Reviewer #1 (Comments for the Author):

The manuscript describes a MALDI-TOF MS method based on multiplex PCR to detection resistance associated mutations in *Candida tropicalis*. The study is relevant but several comments needs to be addressed prior to publication to improve the reproducibility of the results and verify the statements in the introduction and discussion section.

Major Comments:

Line 26: a consistency rate of 95% is mentioned in other parts of the manuscript (line 110) as well. Please elaborate on the meaning of this rate and the percentage itself.

Line 29: the mutations specified are in fact positions in the genes and not mutations, kindly specify the mutations as either DNA positions in the genes like A395T or amino acid positions like Y132F.

Line 40-41: the authors state here and on several other parts of the manuscript that the assay is rapid and low-cost (also in lines 204-205). Kindly elaborate on this in the main text. In regards to low-cost quite some reagents are needed to perform this assay and a MALDI-TOF is needed when compared to microbroth dilution which takes 1 day for yeasts only so also elaborate on the turnaround time aspect.

Line 47: in the used reference 50% of the sepsis cases is not caused by *Candida* species, which seems like a rather high percentage to me. Either add an appropriate reference or adjust the percentage accordingly to a meta review.

Line 51: References to support these statements are missing throughout most of the introduction and discussion, I highly suggest to add many more references to support the text here but on many other occasions as well.

Line 94: table 2 is not present in the files I received, kindly provide table 2 or is this referring to supplementary table 2.

Line 209: the study was only conducted on Chinese DNA samples, were the primers developed on sites that are conserved in a global collection of *C. tropicalis* isolates? If not, then include this a limitation as the primers might not work on isolates that are genetically highly deviant from the east Chinese isolates.

Lines 145-147: here the FKS1 results are mentioned but not further discussed in the discussion section, please do as some echinocandin resistant isolates does not have any FKS1 mutations according to your assay which is interesting and deserved some discussion.

Line 167: over 1 in 10 mutations was not detected with MALDI-TOF according to Sanger results, it seems that heterozygous positions are difficult to detect, elaborate on the discrepancy.

Lines 269-276: this is an exact repetition of lines 238-245, please remove one

Minor Comments

Line 2: SNPs should be written out fully when mentioned first according to abbreviation policies of Microbiology Spectrum

Line 21: replace "correlated to" to "correlated with"

Line 28: rephrase the first part of the sentence as it is incorrect English right now.

Line 36: suggest to remove "huge" as in most regions *C. tropicalis* is not the common pathogen that affects immunocompromised patients mainly.

Line 38: "drastically reduced" depends on the antifungal, I agree on fluconazole but disagree on amphotericin B and echinocandins as resistance is rarely reported, change accordingly.

Line 56: "correctly identify the antifungal susceptibility testing" is incorrect, strains are either identified on species or genus level or evaluated on the AFST, please correct.

Line 104: "These 20 *C. tropicalis*", to what 20 isolates is referred to, 49 isolates were used for the method development and 90 for application so I don't see to what these 20 are referred to.

Line 122: how was the selection done to reach 109 isolates, see the comments to line 104.

Line 144: define mild resistance to fluconazole, was this SDD or provide the MIC

Line 160: "has low throughput" this fully depends on the thermocycler/equipment used so I suggest to rephrase this sentence.

Line 168: an accuracy rate of 100% seems misleading as more than 10% of the mutations is not detected, kindly change accordingly.

Line 178: "Clinical study detected" this is incorrect English, please improve.

Lines 208-209: please specify the limitations more as "further studies" is too vague.

Line 247: From what was DNA extracted, pure cultures or clinical samples?

Line 278: Provide more detail how the PCR and subsequent Sanger sequencing was performed or add a reference.

Reviewer 2

The study titled "A MALDI-TOF MS-based Panel for Detection of Drug Resistance-Associated Multi-SNPs in *Candida tropicalis*" represents a significant contribution in the field of mycology and antifungal resistance detection. The design and execution of this research, including the selection of relevant SNPs and the validation processes, reflect the dedication and expertise of the research team. Their use of MALDI-TOF MS technology for SNP detection is commendable, showcasing an application of this method in the context of antifungal resistance.

The following comments and suggestions may contribute in enhancing the overall quality of the manuscript

Title:

The title is clear and informative.

Abstract:

The abstract summarizes the study well but could benefit from more specific results and numerical data. Include key findings with statistical data (e.g., sensitivity, specificity percentages).

Line 19: *Candida tropicalis* in one of the causes of

Line 27: Give some statistical findings regarding the accuracy and reliability.

Line 28: Avoid starting the sentence with "And"

Lines 29 and 30: Specify the mutations detected not only their positions (like those mentioned in lines 84 and 85).

Importance:

Lines 36 and 37: Specify the patient populations and geographical regions that are most affected by *C. tropicalis* where it represents a public-health challenge.

Line 38: Strengthen the sentence by providing recent global incidence or prevalence rates reported for resistant *C. tropicalis*.
Lines 40 and 41: Determine the methods involved in such comparisons.

Introduction:

The introduction provides a strong background on the significance of *Candida tropicalis* and its resistance to antifungal agents. Inclusion of more recent references to highlight the evolving nature of antifungal resistance and the urgency for new detection methods may be advantageous.

Moreover, the gap in existing methods and the need for a new detection panel is clearly identified. To strengthen this section, consider adding a few more details about the specific limitations of current methods (e.g., sensitivity, specificity).

Line 47: Are the authors sure of this finding (50%). In the reference they cited, it was stated that "In our recently conducted prospective observational study that involved 200 critically ill patients who did not have neutropenia, growth of *Candida* species was observed in blood samples from 17 patients (unpublished data)".

Lines 48 and 49: What about *C. parapsilosis*, and *C. krusei* ?

Line 55: Include recent resistance rates towards the mentioned antifungal classes in China.

Line 56: Use the word "determine" instead of "identify".

Lines 62 and 64: Are broth dilution and/or microbroth dilution assays used in MIC determination, or is disk diffusion used?

Lines 67 to 69: Avoid the redundant part " The risk of mortality after blood samples", It was already highlighted in the former sentence.

Lines 78 to 80: Provide proper reference(s).

Line 85: Provide proper reference(s) and specify the antifungal agent family affected by these mutations.

I suggest adding a paragraph highlighting the most common genes and their corresponding roles in azole and echinocandin resistance to introduce the importance of exploring these genes later in the manuscript.

Results:

Lines 92 to 102: Is it better to include this paragraph in the results section or in the methods section?

This section typically describes the methods, techniques, and materials used in the study. It includes details about how the experiments were conducted, including the establishment and optimization of methods, the design of primers and probes, and any other technical or procedural information necessary to reproduce the experiments. The paragraph provides specific details about the establishment and optimization of a MALDI-TOF MS-based method, primer and probe design, and reaction conditions, all of which fit into the materials and methods description.

On the other hands, results section usually presents the findings of the study, including data, observations, and outcomes derived from the methods described in the materials section. The paragraph does not discuss the findings or results of the method application, but rather the process of setting up and optimizing the method itself.

Therefore, in my point of view the detailed methodological content in the paragraph is appropriate for the "Materials" section.

Lines 98 to 100: The sentence is somewhat ambiguous. It is not entirely clear whether it means that both probes were included in each reaction or that one probe was included in each reaction. It could benefit from clarification. Here is a suggestion that may enhance the clarity of the sentence: "Two different mass probes for the ERG11-428 locus were used, with each reaction including one of the probes, due to the presence of base mutations in some DNA samples."

Lines 103 to 110: It is not clear, there seems to be a discrepancy in the numbers mentioned. The text says 12 mutations were detected, but then it states that most of the mutation sites (32 out of 36) were consistent with Sanger sequencing results. This inconsistency needs to be addressed for clarity.

Lines 108 to 110: The text mentions high reliability and consistency, specifying the statistical methods used to assess this can greatly enhance clarity and rigor. For example, to assess the Intra-

Assay Variability and Inter-Assay Variability, calculations of Coefficient of Variation (CV), Standard Deviation (SD), Intraclass Correlation Coefficient, ...ect may enhance and strength the findings.

TABLE 1 SNPs of 20 *C. tropicalis* (italic) isolates selected for coherence assessment identified by MALDI-TOF MS.

Lines 114 to 119: If available, include specific measures of variability (e.g., standard deviations, coefficients of variation).

Mention any specific steps taken to ensure the accuracy of the replicates. What were the findings of the additional 10 isolates?

What mutations were detected and their relation to antifungal resistance.

Lines 120 to 147: It seems that the authors dealt with the term "mutation" and "locus" equally. Authors should clarify the loci involved and the type of mutations included in their text.

Lines 139 to 141: How many isolates involved per each mutation. Gove number and % per each mutation. Were all the isolates involved also azole resistant ? Line 146: How many isolates? Give number and % if available.

Discussion:

Line 168: How the 100% accuracy was calculated? Is it correct calculation?

The authors did not discuss their findings regarding the echinocandins-related mutations and give possible explanations of the results, comparing such findings with those in literature.

Authors should clearly identify the limitations of the current study.

Methods:

Lines 214 to 216: The distribution of the isolates in this section is confusing with the one stated in the abstract. In this section 49 isolates were used for method validation, while 90 additional isolates were used for clinical research. In the abstract section, 20 isolates were used to determine the methodology, additional 10 for repeatability, and 109 for the clinical research. Avoid discrepancy in presenting the data. Moreover, on what bases the samples were selected and allocated in each category? What

were the inclusion criteria?

Line 226: Refer to the breakpoints or epidemiological cut-off values for each antifungal agent used, and interpret the findings for each isolate as S, SDD, or R in Table S1.

Line 237: Table 2 title: replace "antifungal" with "azoles and echinocandins".

Line 240: It is hard to identify that there are "15" sets of PCR primers in Table S2. There are primers that are used more than once, but with different mass probe. Try to write the primer just once and merge its cell with the other target genes when repeated to avoid confusion.

Line 250: Clarify the protocol used for DNA extraction. (Which method?)

Line 251: Mention the equipment used to assess the purity of the obtained DNA. Was it Nanodrop spectrophotometer ?

Lines 253 and 265: Mention the manufacturer of the PCR and MALDI TOF-MS used in the study. Lines 254 to 264: Make sure of the units used to express volumes (Was it "mL" or "µL"?)

Lines 269 to 276: Avoid redundancy, this paragraph was already mentioned in lines 238 to 245 Line 279: Which 20 isolates were selected? On what basis did the selection take place?

The manuscript describes a MALDI-TOF MS method based on multiplex PCR to detection resistance associated mutations in *Candida tropicalis*. The study is relevant but several comments needs to be addressed prior to publication to improve the reproducibility of the results and verify the statements in the introduction and discussion section.

Major Comments:

Line 26: a consistency rate of 95% is mentioned in other parts of the manuscript (line 110) as well. Please elaborate on the meaning of this rate and the percentage itself.

Line 29: the mutations specified are in fact positions in the genes and not mutations, kindly specify the mutations as either DNA positions in the genes like A395T or amino acid positions like Y132F.

Line 40-41: the authors state here and on several other parts of the manuscript that the assay is rapid and low-cost (also in lines 204-205). Kindly elaborate on this in the main text. In regards to low-cost quite some reagents are needed to perform this assay and a MALDI-TOF is needed when compared to microbroth dilution which takes 1 day for yeasts only so also elaborate on the turnaround time aspect.

Line 47: in the used reference 50% of the sepsis cases is not caused by *Candida* species, which seems like a rather high percentage to me. Either add an appropriate reference or adjust the percentage accordingly to a meta review.

Line 51: References to support these statements are missing throughout most of the introduction and discussion, I highly suggest to add many more references to support the text here but on many other occasions as well.

Line 94: table 2 is not present in the files I received, kindly provide table 2 or is this referring to supplementary table 2.

Line 209: the study was only conducted on Chinese DNA samples, were the primers developed on sites that are conserved in a global collection of *C. tropicalis* isolates? If not, then include this a limitation as the primers might not work on isolates that are genetically highly deviant from the east Chinese isolates.

Lines 145-147: here the FKS1 results are mentioned but not further discussed in the discussion section, please do as some echinocandin resistant isolates does not have any FKS1 mutations according to your assay which is interesting and deserved some discussion.

Line 167: over 1 in 10 mutations was not detected with MALDI-TOF according to Sanger results, it seems that heterozygous positions are difficult to detect, elaborate on the discrepancy.

Lines 269-276: this is an exact repetition of lines 238-245, please remove one

Minor Comments

Line 2: SNPs should be written out fully when mentioned first according to abbreviation policies of Microbiology Spectrum

Line 21: replace "correlated to" to "correlated with"

Line 28: rephrase the first part of the sentence as it is incorrect English right now.

Line 36: suggest to remove "huge" as in most regions *C. tropicalis* is not the common pathogen that affects immunocompromised patients mainly.

Line 38: “drastically reduced” depends on the antifungal, I agree on fluconazole but disagree on amphotericin B and echinocandins as resistance is rarely reported, change accordingly.

Line 56: “correctly identify the antifungal susceptibility testing” is incorrect, strains are either identified on species or genus level or evaluated on the AFST, please correct.

Line 104: “These 20 *C. tropicalis*”, to what 20 isolates is referred to, 49 isolates were used for the method development and 90 for application so I don’t see to what these 20 are referred to.

Line 122: how was the selection done to reach 109 isolates, see the comments to line 104.

Line 144: define mild resistance to fluconazole, was this SDD or provide the MIC

Line 160: “has low throughput” this fully depends on the thermocycler/equipment used so I suggest to rephrase this sentence.

Line 168: an accuracy rate of 100% seems misleading as more than 10% of the mutations is not detected, kindly change accordingly.

Line 178: “Clinical study detected” this is incorrect English, please improve.

Lines 208-209: please specify the limitations more as “further studies” is too vague.

Line 247: From what was DNA extracted, pure cultures or clinical samples?

Line 278: Provide more detail how the PCR and subsequent Sanger sequencing was performed or add a reference.

The study titled "A MALDI-TOF MS-based Panel for Detection of Drug Resistance-Associated Multi-SNPs in *Candida tropicalis*" represents a significant contribution in the field of mycology and antifungal resistance detection. The design and execution of this research, including the selection of relevant SNPs and the validation processes, reflect the dedication and expertise of the research team. Their use of MALDI-TOF MS technology for SNP detection is commendable, showcasing an application of this method in the context of antifungal resistance.

The following comments and suggestions may contribute in enhancing the overall quality of the manuscript

Title:

The title is clear and informative.

Abstract:

The abstract summarizes the study well but could benefit from more specific results and numerical data. Include key findings with statistical data (e.g., sensitivity, specificity percentages).

Line 19: *Candida tropicalis* in one of the causes of

Line 27: Give some statistical findings regarding the accuracy and reliability.

Line 28: Avoid starting the sentence with "And"

Lines 29 and 30: Specify the mutations detected not only their positions (like those mentioned in lines 84 and 85).

Importance:

Lines 36 and 37: Specify the patient populations and geographical regions that are most affected by *C. tropicalis* where it represents a public-health challenge.

Line 38: Strengthen the sentence by providing recent global incidence or prevalence rates reported for resistant *C. tropicalis*.

Lines 40 and 41: Determine the methods involved in such comparisons.

Introduction:

The introduction provides a strong background on the significance of *Candida tropicalis* and its resistance to antifungal agents. Inclusion of more recent references to highlight the evolving nature of antifungal resistance and the urgency for new detection methods may be advantageous.

Moreover, the gap in existing methods and the need for a new detection panel is clearly identified. To strengthen this section, consider adding a few more details about the specific limitations of current methods (e.g., sensitivity, specificity).

Line 47: Are the authors sure of this finding (50%). In the reference they cited, it was stated that "In our recently conducted prospective observational study that involved 200 critically ill patients who did not have neutropenia, growth of candida species was observed in blood samples from 17 patients (unpublished data)".

Lines 48 and 49: What about *C. parapsilosis*, and *C. krusei* ?

Line 55: Include recent resistance rates towards the mentioned antifungal classes in China.

Line 56: Use the word "determine" instead of "identify".

Lines 62 and 64: Are broth dilution and/or microbroth dilution assays used in MIC determination, or is disk diffusion used?

Lines 67 to 69: Avoid the redundant part " The risk of mortality after blood samples", It was already highlighted in the former sentence.

Lines 78 to 80: Provide proper reference(s).

Line 85: Provide proper reference(s) and specify the antifungal agent family affected by these mutations.

I suggest adding a paragraph highlighting the most common genes and their corresponding roles in azole and echinocandin resistance to introduce the importance of exploring these genes later in the manuscript.

Results:

Lines 92 to 102: Is it better to include this paragraph in the results section or in the methods section?

This section typically describes the methods, techniques, and materials used in the study. It includes details about how the experiments were conducted, including the establishment and optimization of methods, the design of primers and probes, and any other technical or procedural information necessary to reproduce the experiments. The paragraph provides specific details about the establishment and optimization of a MALDI-TOF MS-based method, primer and probe design, and reaction conditions, all of which fit into the materials and methods description.

On the other hands, results section usually presents the findings of the study, including data, observations, and outcomes derived from the methods described in the materials section. The paragraph does not discuss the findings or results of the method application, but rather the process of setting up and optimizing the method itself.

Therefore, in my point of view the detailed methodological content in the paragraph is appropriate for the "Materials" section.

Lines 98 to 100: The sentence is somewhat ambiguous. It is not entirely clear whether it means that both probes were included in each reaction or that one probe was included in each reaction. It could benefit from clarification. Here is a suggestion the may enhance the clarity of the sentence: "Two different mass probes for the ERG11-428 locus were used, with each reaction including one of the probes, due to the presence of base mutations in some DNA samples."

Lines 103 to 110: It is not clear, there seems to be a discrepancy in the numbers mentioned. The text says 12 mutations were detected, but then it states that most of the mutation sites (32 out of 36) were consistent with Sanger sequencing results. This inconsistency needs to be addressed for clarity.

Lines 108 to 110: The text mentions high reliability and consistency, specifying the statistical methods used to assess this can greatly enhance clarity and rigor. For example, to assess the Intra-

Assay Variability and Inter-Assay Variability, calculations of Coefficient of Variation (CV), Standard Deviation (SD), Intraclass Correlation Coefficient, ...ect may enhance and strength the findings.

TABLE 1 SNPs of 20 *C. tropicalis* (italic) isolates selected for coherence assessment identified by MALDI-TOF MS.

Lines 114 to 119: If available, include specific measures of variability (e.g., standard deviations, coefficients of variation). Mention any specific steps taken to ensure the accuracy of the replicates. What were the findings of the additional 10 isolates? What mutations were detected and their relation to antifungal resistance.

Lines 120 to 147: It seems that the authors dealt with the term "mutation" and "locus" equally. Authors should clarify the loci involved and the type of mutations included in their text.

Lines 139 to 141: How many isolates involved per each mutation. Gove number and % per each mutation. Were all the isolates involved also azole resistant ?

Line 146: How many isolates? Give number and % if available.

Discussion:

Line 168: How the 100% accuracy was calculated? Is it correct calculation?

The authors did not discuss their findings regarding the echinocandins-related mutations and give possible explanations of the results, comparing such findings with those in literature.

Authors should clearly identify the limitations of the current study.

Methods:

Lines 214 to 216: The distribution of the isolates in this section is confusing with the one stated in the abstract. In this section 49 isolates were used for method validation, while 90 additional isolates were used for clinical research. In the abstract section, 20 isolates were used to determine the methodology, additional 10 for repeatability, and 109 for the clinical research. Avoid discrepancy in presenting the data. Moreover, on what bases the samples were selected and allocated in each category? What were the inclusion criteria?

Line 226: Refer to the breakpoints or epidemiological cut-off values for each antifungal agent used, and interpret the findings for each isolate as S, SDD, or R in Table S1.

Line 237: Table 2 title: replace "antifungal" with "azoles and echicocandins".

Line 240: It is hard to identify that there are "15" sets of PCR primers in Table S2. There are primers that are used onre than once, but with different mass probe. Try to write the primer just once and merge its cell with the other target genes when repeated to avoid confusion.

Line 250: Clarify the protocol used for DNA extraction. (Which method?)

Line 251: Mention the equipment used to assess the purity of the obtained DNA. Was it Nanodrop spectrophotometer ?

Lines 253 and 265: Mention the manufacturer of the PCR and MALDI TOF-MS used in the study.

Lines 254 to 264: Make sure of the units used to express volumes (Was it "mL" or "µL"?)

Lines 269 to 276: Avoid redundancy, this paragraph was already mentioned in lines 238 to 245

Line 279: Which 20 isolates were selected? On what basis did the selection take place?

Dear Editor Rita Oladele and Reviewers,

We are grateful for the thorough review of the manuscript titled “A MALDI-TOF MS-based multiple detection panel of drug resistance-associated multiple single nucleotide polymorphisms in *Candida tropicalis*” (ID: Spectrum00764-24). Thanks very much for taking your time to review this manuscript. We really appreciate all your comments and suggestions. Please find my itemized responses in below and my revisions in the re-submitted files.

Thanks again!

Yours sincerely

Feifei Wan

Reviewer 1:

The manuscript describes a MALDI-TOF MS method based on multiplex PCR to detection resistance associated mutations in *Candida tropicalis*. The study is relevant but several comments need to be addressed prior to publication to improve the reproducibility of the results and verify the statements in the introduction and discussion section.

Major Comments:

1. *Line 26: a consistency rate of 95% is mentioned in other parts of the manuscript (line 110) as well. Please elaborate on the meaning of this rate and the percentage itself.*

Response: Thank you for pointing this out. In fact, 20 *C. tropicalis* isolates were tested across 36 SNP loci, resulting in a total of 720 SNP results (20*36=720). Compared to Sanger sequencing, there were four SNP results that were inconsistent, resulting in an accuracy of 99.44% (716/720). We have made revisions in the abstract (**Line 28**), the main text (**Line 132-133**) and the discussion (**Line 199**).

2. *Line 29: the mutations specified are in fact positions in the genes and not mutations, kindly specify the mutations as either DNA positions in the genes like A395T or amino acid positions like Y132F.*

Response: Thank you for pointing this out. The mutations are now presented as nucleotide changes at the respective positions within the genes. **(Line 29-31)**

3. *Line 40-41: the authors state here and on several other parts of the manuscript that the assay is rapid and low-cost (also in lines 204-205). Kindly elaborate on this in the main text. In regards to lowcost quite some reagents are needed to perform this assay and a MALDI-TOF is needed when compared to microbroth dilution which takes 1 day for yeasts only so also elaborate on the turnaround time aspect.*

Response: Thank you for your comment. We would like to clarify that the assay requires only 6-8 hours for completion, making it relatively rapid. **(Line 196)** Regarding the term "low-cost," we meant it in comparison to other molecular biology methods that typically involve higher expenses **(Line 94-95)**.

4. *Line 47: in the used reference 50% of the sepsis cases is not caused by Candida species, which seems like a rather high percentage to me. Either add an appropriate reference or adjust the percentage accordingly to a meta review.*

Response: Thank you for your suggestion. We have revised the text to clarify that approximately 50% of candidemia cases occur in ICU. **(Line 53)** Additionally, we have included appropriate references to support this statement **(Reference 1,2)**.

5. *Line 51: References to support these statements are missing throughout most of the introduction and discussion, I highly suggest to add many more references to support the text here but on many other occasions as well.*

Response: Thank you for your suggestion. We have reviewed the introduction and discussion sections and added several references to support the statements. **(Reference 1-17, 21-24, 26-29)**

6. *Line 94: table 2 is not present in the files I received, kindly provide table 2 or is this referring to supplementary table 2.*

Response: Thank you for your comment. Table 2 is now located on **line 290-291**. And the table has been renamed **Table 3. (Line 290-291)**

7. *Line 209: the study was only conducted on Chinese DNA samples, were the primers developed on sites that are conserved in a global collection of C. tropicalis isolates? If not, then include this a limitation as the primers might not work on isolates that are genetically highly deviant from the east Chinese isolates.*

Response: Thank you for your suggestion. The primers used in this study were designed based on the reference sequences of *Candida tropicalis* downloaded from the NCBI database. Although our study was conducted using Chinese DNA samples, the primers are applicable more broadly.

8. *Lines 145-147: here the FKS1 results are mentioned but not further discussed in the discussion section, please do as some echinocandin resistant isolates does not have any FKS1 mutations according to your assay which is interesting and deserved some discussion.*

Response: Thank you for your suggestion. We have added a discussion on the *FKS1* results in the discussion. **(Line 228-241)**

9. *Line 167: over 1 in 10 mutations was not detected with MALDI-TOF according to Sanger results, it seems that heterozygous positions are difficult to detect, elaborate on the discrepancy.*

Response: Thank you for your suggestion. In fact, 20 *C. tropicalis* isolates were tested across 36 SNP loci, resulting in a total of 720 SNP results (20*36=720). Compared to Sanger sequencing, there were four SNP results that were inconsistent, resulting in an accuracy of 99.44% (716/720). **(Line 132-133)** However, it is true that at these four loci, the distinction between heterozygous and non-mutations positions proved difficult. This issue has been addressed and discussed in the discussion section **(Line 248-251)**.

10. *Lines 269-276: this is an exact repetition of lines 238-245, please remove one.*

Response: Thank you for pointing this out. We have deleted that latter part of the repetitive content to avoid redundancy.

Minor Comments

1. *Line 2: SNPs should be written out fully when mentioned first according to abbreviation policies of Microbiology Spectrum.*

Response: Thank you for pointing this out. In accordance with the abbreviation policies of Microbiology Spectrum, I have revised the manuscript to write out "Single Nucleotide Polymorphisms (SNPs)" in full. **(Line2)**

2. *Line 21: replace “correlated to” to “correlated with”*

Response: Thank you for pointing this out. We have replaced "correlated to" with "correlated with". **(Line 22)**

3. *Line 28: rephrase the first part of the sentence as it is incorrect English right now.*

Response: Thank you for your suggestion. We have revised the sentence. **(Line 28)**

4. *Line 36: suggest to remove “huge” as in most regions C. tropicalis is not the common pathogen that affects immunocompromised patients mainly.*

Response: Thank you for your suggestion. We have removed the word "huge". **(Line 36)**

5. *Line 38: “drastically reduced” depends on the antifungal, I agree on fluconazole but disagree on amphotericin B and echinocandins as resistance is rarely reported, change accordingly.*

Response: Thank you for your suggestion. The azole resistance rate of *C. tropicalis* ranges from 0% to 30%. Between 2009 and 2018, the China Hospital Invasive Fungal Surveillance Network reported an increase in fluconazole and voriconazole resistance from 5.7% to ~30%. Although resistance to echinocandins and amphotericin B remains low, multi-resistance to echinocandins and azoles has been observed. The above content has been added to the main text. **(Line 39-42)**

6. *Line 56: “correctly identify the antifungal susceptibility testing” is incorrect, strains are either identified on species or genus level or evaluated on the AFST, please correct.*

Response: Thank you for your suggestion. We agree that "evaluated the AFST" is the correct phrasing. However, based on another reviewer's suggestion, we have added a description of the molecular mechanisms of *C. tropicalis* resistance in a subsequent section. To ensure a smooth transition between these topics, I have removed this sentence.

7. *Line 104: “These 20 C. tropicalis”, to what 20 isolates is referred to, 49 isolates were used for the method development and 90 for application so I don’t see to what these 20 are referred to.*

Response: Thank you for pointing this out. We apologize for the oversight. Actually, to evaluate the repeatability of the method, 10 isolates of *C. tropicalis* from different years and with varying antifungal resistance phenotypes were selected (Table S1), ensuring a balanced representation of resistance profiles. Additionally, 20 isolates were chosen to assess consistency, also derived from diverse years and resistance phenotypes, maximizing the detection of mutations among the included SNP loci. To evaluate the clinical performance of the method, all *C. tropicalis* isolates collected between 2017 and 2021 by the ECIFIG were included. The specific selection criteria and numbers have been detailed in the materials and methods. **(Line 272-278)**

8. *Line 122: how was the selection done to reach 109 isolates, see the comments to line 104.*

Response: Thank you for your suggestion. The 109 *C. tropicalis* isolates were collected from 2017 to 2021 by the Eastern China Invasive Fungal Infection Group, representing all the strains of *C. tropicalis* gathered during that period. **(Line 272-278)**

9. *Line 144: define mild resistance to fluconazole, was this SDD or provide the MIC.*

Response: Thank you for your suggestion. We have clarified the resistance level by providing the MIC value. **(Line 169)**

10. *Line 160: “has low throughput” this fully depends on the thermocycler/equipment used so I suggest to rephrase this sentence.*

Response: Thank you for your suggestion. The sentence has been revised to the following: PCR detection sensitivity for SNP depends on the equipment. **(Line 186-187)**

11. *Line 168: an accuracy rate of 100% seems misleading as more than 10% of the mutations is not detected, kindly change accordingly.*

Response: Thank you for your suggestion. In fact, 20 *C. tropicalis* isolates were tested across 36 SNP loci, resulting in a total of 720 SNP results (20*36=720). Compared to Sanger sequencing,

there were four SNP results that were inconsistent, resulting in an accuracy of 99.44% (716/720).

We have made revisions in the abstract (Line 28), the main text (Line 132-133) and the discussion. (Line 199)

12. *Line 178: "Clinical study detected" this is incorrect English, please improve.*

Response: Thank you for pointing this out. The sentence was already revised **(Line 200)**

13. *Lines 208-209: please specify the limitations more as "further studies" is too vague.*

Response: Thanks for your suggestion. The study has some limitations. During the performance validation of the panel, it was discovered that due to base mismatches, the panel may incorrectly identify non-mutation as heterozygous mutation and vice versa, which could result in the misinterpretation of results. Given the diverse mechanisms of drug resistance in *C. tropicalis*, this panel currently only includes the most significant mutations, potentially overlooking less common or emerging resistance mechanisms. Additionally, the panel is designed to detect mutations from pure cultures of *C. tropicalis*, which may limit direct application in clinical settings. Further efforts will be required to adapt this method for detecting SNPs directly from clinical specimens, such as blood or tissue samples, to enhance its clinical utility. The above content has been added to the main text. **(Line 248-257)**

14. *Line 247: From what was DNA extracted, pure cultures or clinical samples?*

Response: Thanks for your question. The DNA used in this study was extracted from pure cultures. The detailed extraction method has been added to the materials and methods section.

(Line 303-314)

15. *Line 278: Provide more detail how the PCR and subsequent Sanger sequencing was performed or add a reference.*

Response: Thanks for your suggestion. The detail of how the PCR and subsequent Sanger sequencing were performed have been added to the materials and methods section. **(Line 361-367)**

Reviewer 2:

The study titled "A MALDI-TOF MS-based Panel for Detection of Drug Resistance-Associated Multi-SNPs in *Candida tropicalis*" represents a significant contribution in the field of mycology and antifungal resistance detection. The design and execution of this research, including the selection of relevant SNPs and the validation processes, reflect the dedication and expertise of the research team. Their use of MALDI-TOF MS technology for SNP detection is commendable, showcasing an application of this method in the context of antifungal resistance.

The following comments and suggestions may contribute in enhancing the overall quality of the manuscript.

Title:

The title is clear and informative.

Abstract:

1. *The abstract summarizes the study well but could benefit from more specific results and numerical data. Include key findings with statistical data (e.g., sensitivity, specificity percentages).*

Response: Thanks for your suggestion. We have included key findings using accuracy and repeatability: Ten isolates were selected to test repeatability, and another 20 isolates of *C. tropicalis* were selected to validate consistency. The both intra-assay and inter-assay repeatability of the panel were 100%, with the loci accuracy 99.44% (716/720). **(Line 27-28)**

2. *Line 19: Candida tropicalis in one of the causes of*

Response: Thanks for your suggestion. We have revised the sentence to the following: "*Candida tropicalis* is one of the main causes of invasive candidiasis." **(Line 20)**

3. *Line 27: Give some statistical findings regarding the accuracy and reliability.*

Response: Thanks for your suggestion. To confirm reliability, we performed both intra-assay and inter-assay repeatability testing, with both achieving 100% repeatability. Additionally, to ensure accuracy, we calculated the loci accuracy rate, which was 99.44%. **(Line 27-28)**

4. *Line 28: Avoid starting the sentence with "And"*

Response: Thank you for your suggestion. We have revised the sentence. **(Line 28)**

5. *Lines 29 and 30: Specify the mutations detected not only their positions (like those mentioned in lines 84 and 85).*

Response: Thank you for your suggestion. I have revised the sentence to specify the detected mutations rather than just their positions. **(Line 29-33)**

Importance:

1. *Lines 36 and 37: Specify the patient populations and geographical regions that are most affected by *C. tropicalis* where it represents a public-health challenge.*

Response: Thanks for your suggestion. *C. tropicalis* is one of the top four *Candida* species responsible for candidiasis, particularly in the Asia-Pacific region and Latin America, notably affecting patients with neutropenia and malignancies. The above content has been added into the main text. **(Line 36-39)**

2. *Line 38: Strength the sentence by providing recent global incidence or prevalence rates reported for resistant *C. tropicalis*.*

Response: Thank you for your suggestion. The azole resistance rate of *C. tropicalis* ranges from 0% to 30%. Between 2009 and 2018, the China Hospital Invasive Fungal Surveillance Network reported an increase in fluconazole and voriconazole resistance from 5.7% to ~30%. Although resistance to echinocandins and amphotericin B remains low, multi-resistance to echinocandins and azoles has been observed. The above content has been added to the main text. **(Line 39-43)**

3. *Lines 40 and 41: Determine the methods involved in such comparisons.*

Response: Thank you for suggestion. The methods involved in such comparisons have been clearly specified, and their respective limitations have also been briefly described. **(Line 44-46)**

Introduction:

1. *The introduction provides a strong background on the significance of Candida tropicalis and its resistance to antifungal agents. Inclusion of more recent references to highlight the evolving nature of antifungal resistance and the urgency for new detection methods may be advantageous.*

Moreover, the gap in existing methods and the need for a new detection panel is clearly identified. To strengthen this section, consider adding a few more details about the specific limitations of current methods (e.g., sensitivity, specificity).

Response: Thank you for your suggestion. We have included more recent references (**references 6-13**) to highlight the evolving nature of antifungal resistance and emphasized that the emergence of azole resistance has led to increased exposure to echinocandins. (**Line 58-63**) Moreover, we have expanded the introduction to include details on the gap in existing methods (**Line 76-85**).

2. *Line 47: Are the authors sure of this finding (50%). In the reference they cited, it was stated that "In our recently conducted prospective observational study that involved 200 critically ill patients who did not have neutropenia, growth of candida species was observed in blood samples from 17 patients (unpublished data)".*

Response: Thank you for pointing this out. We apologize for the incorrect citation and have included appropriate references (**references 1,2**) to support this statement. Our intention was to convey that approximately 50% of candidemia cases occur in the ICU. (**Line 51-53**)

3. *Lines 48 and 49: What about C. parapsilosis, and C. krusei ?*

Response: Thank you for pointing this out. We have revised our statement to acknowledge that while more than 15 distinct *Candida spp.* can cause human infections, the six most clinically relevant are *Candida albicans*, *Candida parapsilosis*, *Pichia kudriavzevii*, *Nakaseomyces glabratus*, *Candida tropicalis*, and *Candida auris*. (**Line 53-56**)

4. *Line 55: Include recent resistance rates towards the mentioned antifungal classes in China.*

Response: Thank you for pointing this out. Recent evidence indicates an azole resistance rate of 20-50% for *C. tropicalis* in the Asia-Pacific region, India, Turkey and Algeria. The emergence of fluconazole resistance has leads to increased echinocandin exposure, with *C. tropicalis* isolates

resistant to both azoles and echinocandins being reported in the USA, India, Taiwan and mainland China, although the prevalence remains low (<2%). **(Line 58-63)**

5. *Line 56: Use the word "determine" instead of "identify".*

Response: Thank you for pointing this out. We agree that "determine the antifungal susceptibility testing" is the correct phrasing. However, we have added a description of the molecular mechanisms of *C. tropicalis* resistance in a subsequent section. To ensure a smooth transition between these topics, I have removed this sentence.

6. *Lines 62 and 64: Are broth dilution and/or microbroth dilution assays used in MIC determination, or is disk diffusion used?*

Response: Thank you for pointing this out. The Clinical and Laboratory Standards Institute (CLSI) and the European Committee on Antimicrobial Susceptibility Testing (EUCAST) recommend the microbroth dilution method for determining the minimum inhibitory concentration (MIC). In our study, we utilized the Sensititre YeastOne™ panel for detecting the MIC of *Candida tropicalis*, which is a commercially available product developed based on the microbroth dilution method. **(Line 76)**

7. *Lines 67 to 69: Avoid the redundant part " The risk of mortality after blood samples", It was already highlighted in the former sentence.*

Response: Thank you for pointing this out. We have removed the redundant section. However, we have streamlined the introduction in this section to focus more on the molecular biology methods.

8. *Lines 78 to 80: Provide proper reference(s).*

Response: Thank you for your suggestion. We have provided proper references. **(Line 91-93)**
(Reference 21-24)

9. *Line 85: Provide proper reference(s) and specify the antifungal agent family affected by these mutations. I suggest adding a paragraph highlighting the most common genes and their corresponding roles in azole and echinocandin resistance to introduce the importance of exploring these genes later in the manuscript.*

Response: Thank you for your suggestion. Azoles and echinocandins are the first-line therapeutic agents for IC. Azoles targeting 14-alpha-demethylase (Erg11) inhibit the ergosterol biosynthesis pathway in fungal cell membranes, whereas echinocandins bind to the Fks p-subunit of the β -(1,3)-D-glucan synthase complex, thereby blocking synthesis of β -(1,3)-D-glucan. Azole resistance can develop through multiple mechanisms, including substitutions in Erg11, which reduce the drug-binding affinity between ergosterol and azoles. Additionally, overexpression of Erg11, driven by gain-of-function mutations to the transcriptional activator *UPC2*, can lead to overexpression of drug targets. Azole resistance is also associated with upregulation of drug efflux pumps. Mutations to *TAC1* or *MRR1* lead to the upregulation of adenosine triphosphate-binding cassette transporter *CDR1* and the major facilitator superfamily transporter *MDR1*. Resistance to echinocandins primarily involves mutations to *FKSI*. The above content has been added to the main text. **(Line 64-75)**

Results:

1. *Lines 92 to 102: Is it better to include this paragraph in the results section or in the methods section?*

This section typically describes the methods, techniques, and materials used in the study. It includes details about how the experiments were conducted, including the establishment and optimization of methods, the design of primers and probes, and any other technical or procedural information necessary to reproduce the experiments. The paragraph provides specific details about the establishment and optimization of a MALDI-TOF MS-based method, primer and probe design, and reaction conditions, all of which fit into the materials and methods description.

On the other hands, results section usually presents the findings of the study, including data, observations, and outcomes derived from the methods described in the materials section. The paragraph does not discuss the findings or results of the method application, but rather the process of setting up and optimizing the method itself.

Therefore, in my point of view the detailed methodological content in the paragraph is appropriate for the "Materials" section.

Response: Thank you for your suggestion. We have removed the detailed methodological content in the paragraph to the "Materials and Method" section. **(Line 336-347)**

2. *Lines 98 to 100: The sentence is somewhat ambiguous. It is not entirely clear whether it means that both probes were included in each reaction or that one probe was included in each reaction. It could benefit from clarification. Here is a suggestion may enhance the clarity of the sentence: "Two different mass probes for the ERG11-428 locus were used, with each reaction including one of the probes, due to the presence of base mutations in some DNA samples."*

Response: Thank you for your suggestion. We have revised the sentence. **(Line 343-345)**

3. *Lines 103 to 110: It is not clear, there seems to be a discrepancy in the numbers mentioned. The text says 12 mutations were detected, but then it states that most of the mutation sites (32 out of 36) were consistent with Sanger sequencing results. This inconsistency needs to be addressed for clarity.*

Response: Thank you for your suggestion. In fact, 20 *C. tropicalis* isolates were tested across 36 SNP loci, resulting in a total of 720 SNP results ($20 \times 36 = 720$). Compared to Sanger sequencing, there were four SNP results that were inconsistent, resulting in an accuracy of 99.44% (716/720). We have made revisions in the abstract **(Line 28)**, the main text **(Line 132-133)** and the discussion **(Line 199)**.

4. *Lines 108 to 110: The text mentions high reliability and consistency, specifying the statistical methods used to assess this can greatly enhance clarity and rigor. For example, to assess the Intra-Assay Variability and Inter-Assay Variability, calculations of Coefficient of Variation (CV), Standard Deviation (SD), Intraclass Correlation Coefficient, ...ect may enhance and strength the findings.*

Response: Thank you for your suggestion. Since our study is based on qualitative analysis, and both inter-assay and intra-assay results were consistent, it is challenging to use coefficients of

variation or standard deviations to describe variability. Therefore, we have only described that the panel exhibited 100% repeatability. **(Line 101-105)**

5. *TABLE 1 SNPs of 20 C. tropicalis (italic) isolates selected for coherence assessment identified by MALDI-TOF MS.*

Response: Thank you for pointing out the formatting issue. We apologize for the oversight regarding the italicization of the word. We have corrected it in the revised manuscript. **(Line 136-137)**

6. *Lines 114 to 119: If available, include specific measures of variability (e.g., standard deviations, coefficients of variation). Mention any specific steps taken to ensure the accuracy of the replicates. What were the findings of the additional 10 isolates? What mutations were detected and their relation to antifungal resistance.*

Response: Thank you for your suggestion. Since our study is based on qualitative analysis, and both inter-assay and intra-assay results were consistent, it is challenging to use coefficients of variation or standard deviations to describe variability. Therefore, we have only described that the panel exhibited 100% repeatability. Additionally, in the repeatability evaluation, we selected isolates with known drug susceptibility: 4 isolates susceptible to fluconazole, 1 isolate susceptible dose-dependent to fluconazole, and 5 isolates resistant to fluconazole, for a total of 10 isolates. The repeatability for these isolates was completely consistent, and detailed results are described in the main text. **(Line 106-120)**

7. *Lines 120 to 147: It seems that the authors dealt with the term "mutation" and "locus" equally. Authors should clarify the loci involved and the type of mutations included in their text.*

Response: Thank you for your suggestion. In the revised manuscript, we have clarified the term "mutation" and "locus" in the main text. **(Line 140-175)**

8. *Lines 139 to 141: How many isolates involved per each mutation. Gove number and % per each mutation. Were all the isolates involved also azole resistant?*

Response: Thank you for your suggestion. We have added details regarding the number of isolates involved for each mutation, along with the corresponding percentages. Additionally, we

have clarified all isolates with mutations were azole resistant in the revised manuscript. The original sentence has been revised to the following: However, not all azole-resistant isolates carried these three mutations, as some carried mutations at other loci associated with azole resistance, such as *UPC2-1029* (2.75%, 3/109), *UPC2-751* (65.14%, 71/109), *TAC1-491* (62.39%, 68/109), *ERG11-433* (1.83%, 2/109), and *UPC2-866* (64.22%, 70/109). **(Line 162-163)**

9. *Line 146: How many isolates? Give number and % if available.*

Response: Thank you for your suggestion. We have included the number of isolates and the corresponding percentages in the revised manuscript. **(Line 170-171)**

Discussion:

1. *Line 168: How the 100% accuracy was calculated? Is it correct calculation? The authors did not discuss their findings regarding the echinocandins-related mutations and give possible explanations of the results, comparing such findings with those in literature.*

Response: Thank you for pointing this out. In fact, 20 *C. tropicalis* isolates were tested across 36 SNP loci, resulting in a total of 720 SNP results (20*36=720). Compared to Sanger sequencing, there were four SNP results that were inconsistent, resulting in an accuracy of 99.44% (716/720). We have made revisions in in the abstract **(Line 28)**, the main text **(Line 132-133)** and the discussion. **(Line 199)**. Additionally, we have expanded the discussion section to include an analysis of echinocandin-related mutations and to compare our findings with relevant literature. **(Line 230-243)**

2. *Authors should clearly identify the limitations of the current study.*

Response: Thank you for your suggestion. The study has some limitations. During the performance validation of the panel, it was discovered that due to base mismatches, the panel may incorrectly identify non-mutation as heterozygous mutation and vice versa, which could result in the misinterpretation of results. Given the diverse mechanisms of drug resistance in *C. tropicalis*, this panel currently only includes the most significant mutations, potentially overlooking less common or emerging resistance mechanisms. Additionally, the panel is designed to detect mutations from pure cultures of *C. tropicalis*, which may limit direct application in clinical

settings. Further efforts will be required to adapt this method for detecting SNPs directly from clinical specimens, such as blood or tissue samples, to enhance its clinical utility. The above content has been added to the main text to clearly identify the limitations of the current study.

(Line 248-257)

Methods:

1. *Lines 214 to 216: The distribution of the isolates in this section is confusing with the one stated in the abstract. In this section, 49 isolates were used for method validation, while 90 additional isolates were used for clinical research. In the abstract section, 20 isolates were used to determine the methodology, additional 10 for repeatability, and 109 for the clinical research. Avoid discrepancy in presenting the data. Moreover, on what bases the samples were selected and allocated in each category? What were the inclusion criteria?*

Response: Thank you for pointing this out. In fact, we used 20 isolates for method validation, an additional 10 for repeatability, and 109 for clinical performance evaluation. We have also added specific details regarding the basis for sample selection and allocation in the 'Materials and Methods' section. **(Line 272-278)**

2. *Line 226: Refer to the breakpoints or epidemiological cut-off values for each antifungal agent used, and interpret the findings for each isolate as S, SDD, or R in Table S1.*

Response: Thank you for your suggestion. We have interpreted the findings for each isolate as susceptible (S), susceptible dose-dependent (SDD), or resistant (R) in Table S1.

3. *Line 237: Table 2 title: replace "antifungal" with "azoles and echinocandins".*

Response: Thank you for your suggestion. We have revised the sentence. **(Line 289-299)**

4. *Line 240: It is hard to identify that there are "15" sets of PCR primers in Table S2. There are primers that are used more than once, but with different mass probe. Try to write the primer just once and merge its cell with the other target genes when repeated to avoid confusion.*

Response: Thank you for your suggestion. We have revised Table S2 to list each primer only once and merged cells for any repeated primers.

5. *Line 250: Clarify the protocol used for DNA extraction. (Which method?)*

Response: Thank you for your suggestion. We used the phenol-chloroform extraction method for DNA extraction, and the detailed protocol has been described in the manuscript. **(Line 303-315)**

6. *Line 251: Mention the equipment used to assess the purity of the obtained DNA. Was it Nanodrop spectrophotometer?*

Response: Thank you for your suggestion. We have revised the manuscript to specify that DNA concentration and purity were measured using a Quawell Q5000 Micro-Volume UV-Vis Spectrophotometer (Quawell Technology, Sunnyvale, CA, USA). **(Line 315-316)**

7. *Lines 253 and 265: Mention the manufacturer of the PCR and MALDI TOF-MS used in the study.*

Response: Thank you for your suggestion. We have added the manufacturers of the PCR equipment and MALDI-TOF MS used in the study to the revised manuscript.

8. *Lines 254 to 264: Make sure of the units used to express volumes (Was it "mL" or "µL"?)*

Response: Thank you for your suggestion. We have corrected the incorrect 'mL' to 'µL' in the revised manuscript. **(Line 299-335)**

9. *Lines 269 to 276: Avoid redundancy, this paragraph was already mentioned in lines 238 to 245.*

Response: Thank you for your suggestion. We have removed the redundancy.

10. *Line 279: Which 20 isolates were selected? On what basis did the selection take place?*

Response: Thank you for your suggestion. We have revised the text to specify that 20 *C. tropicalis* isolates from different years and with varying antifungal resistance phenotypes were selected (see Table S1). We employed stratified sampling to ensure a balanced representation of antifungal resistance phenotypes while maximizing the detection of the included SNP sites. **(Line 272-278)**

Re: Spectrum00764-24R1 (**A MALDI-TOF MS-based multiple detection panel of drug resistance-associated single nucleotide polymorphisms in *Candida tropicalis***)

Dear Prof. WenJuan Wu:

Your manuscript has been accepted, and I am forwarding it to the ASM production staff for publication. Your paper will first be checked to make sure all elements meet the technical requirements. ASM staff will contact you if anything needs to be revised before copyediting and production can begin. Otherwise, you will be notified when your proofs are ready to be viewed.

Sincerely,
Rita Oladele
Editor
Microbiology Spectrum

Reviewer #1 (Comments for the Author):

The comments of both reviewers have been incorporated appropriately in the revised manuscript and I have no further comments as such.